# Extranodal Extension Predicts Poor Survival Outcomes among Patients with Bladder Cancer

**DOI:** 10.3390/cancers13164108

**Published:** 2021-08-15

**Authors:** Yi-An Liao, Chun-Ju Chiang, Wen-Chung Lee, Bo-Zhi Zhuang, Chung-Hsin Chen, Yeong-Shiau Pu

**Affiliations:** 1Department of Urology, National Taiwan University Hospital, Taipei 10002, Taiwan; tomchad321@ntuh.gov.tw (Y.-A.L.); yspu@ntu.edu.tw (Y.-S.P.); 2Institute of Epidemiology and Preventive Medicine, College of Public Health, National Taiwan University, Taipei 10055, Taiwan; ruru.chiang@cph.ntu.edu.tw (C.-J.C.); wenchung@ntu.edu.tw (W.-C.L.); augustine@cph.ntu.edu.tw (B.-Z.Z.); 3Taiwan Cancer Registry, Taipei 10055, Taiwan

**Keywords:** bladder cancer, extranodal extension, adjuvant chemotherapy, overall survival, cancer-specific survival

## Abstract

**Simple Summary:**

Several lymph node-related prognosticators have been reported in bladder cancer patients with lymph node involvement who undergo radical cystectomy. However, the role of extranodal extension (ENE) remains debatable for outcome prediction. The aim of our study is to investigate the association between ENE and prognosis in Taiwanese patients with pathological nodal bladder cancer who were treated with radical cystectomy using a nationwide database. Our study concluded that ENE significantly reduced OS and CSS among the pathological nodal bladder cancer patients. After the identification of pathological nodal disease, adjuvant chemotherapy was associated with better survival outcomes in the patients with ENE.

**Abstract:**

Background: Several lymph node-related prognosticators were reported in bladder cancer patients with lymph node involvement and receiving radical cystectomy. However, extranodal extension (ENE) remained a debate to predict outcomes. Methods: A retrospective analysis of 1303 bladder cancer patients receiving radical cystectomy and bilateral pelvic lymph node dissection were identified in the National Taiwan Cancer Registry database from 2011 to 2017. Based on the 304 patients with lymph node involvement, the presence of ENE and major clinical information were recorded and calculated. The overall survival (OS) and cancer-specific survival (CSS) were estimated with Kaplan–Meier analysis and compared using the log-rank test. Hazard ratios (HR) and the associated 95% confidence intervals were calculated in the univariate and stepwise multivariable models. Results: In the multivariable analysis, ENE significantly reduced OS (HR = 1.74, 95% CI 1.09–2.78) and CSS (HR = 1.69, 95% CI 1.01–2.83) more than non-ENE. In contrast, adjuvant chemotherapy was significantly associated with better OS and CSS upon the identification of pathological nodal disease. Conclusions: Reduced OS and CSS outcomes were observed in the pathological nodal bladder cancer patients with ENE compared with those without ENE. After the identification of pathological nodal disease, adjuvant chemotherapy was associated with better survival outcomes.

## 1. Introduction

Bladder cancer is the most common genitourinary malignancy. According to the World Health Organization, 573,000 new cases of bladder cancer and 213,000 deaths related to bladder cancer were reported worldwide in 2020 [1]. A radical cystectomy with bilateral pelvic lymph node dissection is the standard treatment for patients with muscle-invasive bladder cancer (MIBC) [2]. However, the 10-year mortality rate still reaches 80% in cases with pathological pelvic lymph node involvement [3]. By contrast, some patients with MIBC show favorable survival rates following radical cystectomy despite confirmed pathological involvement of lymph nodes. Based on the current evidence, the reported prognosticators include the pathological stage [4] and lymphovascular invasion [5] of the primary tumor, the pathological nodal stage, the number of lymph nodes involved [6], the number of lymph nodes removed [7], and lymph node density (LND) [6]. However, the role of extranodal extension (ENE) of the involved lymph nodes in the prediction of patients’ overall survival (OS) and cancer-specific survival (CSS) remains debatable.

Fajkovic retrospectively analyzed 748 patients with a high risk of noninvasive bladder cancer or MIBC who were treated using radical cystectomy and lymphadenectomy and concluded that ENE increased disease recurrence and cancer-specific mortality rates [8]. Fleischmann analyzed the outcomes of 124 similar patients with MIBC who were treated using radical cystectomy and standardized extended bilateral pelvic lymphadenectomy and reported that extracapsular extension of the lymph nodes significantly worsened the patients’ outcomes [5]. Nevertheless, some authors considered that the significance of ENE as a prognostic factor is controversial [9]. A large prospective study is required to determine the prognostic role of ENE.

To eliminate the debate on the prognostic value of ENE in patients with MIBC, we investigated the association between ENE and prognosis in Taiwanese patients with MIBC and lymph node involvement who were treated with radical cystectomy using a nationwide database. We observed that ENE reduced the OS and CSS among patients with MIBC. We also found that adjuvant chemotherapy seemed to help some patients with MIBC and ENE.

## 2. Materials and Methods

### 2.1. Patient Selection from Taiwan Cancer Registry

The National Taiwan Cancer Registry (TCR) is a population-based cancer registry system established in 1979 [10]. Owing to the high data quality and completeness of the database, the TCR is one of the highest-quality cancer registries worldwide [11]. Since 2002, the TCR established a long-form database with detailed information on cancer staging, treatment, and recurrence. Further, in 2011, the long-form database started collecting information regarding smoking habits, body mass index (BMI), and cancer site-specific factors such as information regarding ENE among patients with bladder cancer. In this study, all patients who underwent radical cystectomy for diagnosis of urothelial carcinoma (UC) (ICD-O topography code C67 and morphology codes 8050, 8120–8124, and 8130–8131) and lymph node metastasis between 2011 and 2017 were selected from the TCR. In Taiwan, tumor topography, morphology, and grade are coded according to the International Classification for Disease Oncology 3rd Edition (ICD-O-3) [12]. To classify cancer stage, the seventh edition of the American Joint Committee on Cancer TNM classification system was used during the study period [13].

### 2.2. Data Collection

Clinical data for analysis included patient age (25–64, 65–74, and ≥75 years), sex, tumor grade (low and high grade), pathological T and N stage, tumor size (<4 and ≥4 cm), surgical margin (free or not), ENE, LND, treatment pattern, smoking habits, and BMI. All lymphoid tissues excised from pelvic lymph node dissections were submitted for histological examination. LND was defined as the ratio of the number of positive lymph nodes to the total number of lymph nodes removed. Treatment was categorized into four groups: no perioperative chemotherapy, neo-adjuvant chemotherapy, adjuvant chemotherapy, and neo-adjuvant plus adjuvant chemotherapy. This study was approved by the National Taiwan University Hospital Research Ethics Committee (201801116RINA and 201912201W).

Survival duration was defined as the time from the date of initial diagnosis to the date of death or until the last date of follow-up. The vital status of patients was evaluated using the national death certificate database maintained by the Department of Statistics, Ministry of Health and Welfare, Taiwan, and collected up to 31 December 2019. The records of patients whose date of death was unknown were excluded.

### 2.3. Statistical Analysis

The chi-squared test was used to evaluate the association between categorical variables. The OS and CSS were estimated using Kaplan–Meier analysis and compared using the log-rank test. The association between clinicopathological variables and outcomes was assessed using Cox proportional-hazards regression models. Hazard ratios (HRs) and the associated 95% confidence intervals (CIs) were calculated in the univariate and stepwise multivariable models. Two-sided statistical significance was defined as *p* < 0.05. All analyses were performed using SAS version 9.4 statistical software (SAS Institute, Cary, NC, USA).

## 3. Results

### 3.1. Demographics

From 2011 to 2017, a total of 1303 patients with bladder cancer who had undergone radical cystectomy were identified in the TCR database. Among them, 304 (18.9%) patients with lymph node involvement were included in this study. (Table 1) The median age of this cohort (70.4% male) was 66 (range, 25–88 years) years. The pathological samples revealed the proportions of pT0, pTa/1/is, pT2, pT3, pT4 and positive surgical margins as 3.0%, 2.0%, 15.5%, 51.3%, 28.3%, and 16.8%, respectively. Approximately 46.7% of the samples showed an LND of ≥20%. In total, 48, 75, and 181 patients had positive, negative, and unknown ENE status, respectively. Patients with ENE had a higher rate of multiple primaries (93.8% vs. 72.0%) and LND ≥ 20% (60.4% vs. 37.3%) than patients without ENE. In addition, more patients with ENE received perioperative chemotherapy than those without ENE (81.2% vs. 57.3%). However, the proportion of patients who received adjuvant chemotherapy alone was similar between the ENE and non-ENE groups (45.8% vs. 41.3%).

### 3.2. Overall Survival

Until December 2019, a total of 212 (69.7%) patients died of any reason. Compared with patients without ENE, those with ENE and unknown status of ENE had significantly lower survival rates (Figure 1, log-rank test *p* = 0.044). The 5-year OS rates of patients with ENE, unknown ENE status, and without ENE were 19.1%, 24.9%, and 37.2%, respectively. In addition to ENE status, age, pathological T stage, surgical margin, tumor size, perioperative chemotherapy, and current smoking habits were associated with OS in the univariable analysis (Table 2). Multivariable analysis revealed that age, pathological T stage, perioperative chemotherapy, ENE status, tumor size, and surgical margin were significant prognostic factors. Compared with patients without ENE, those with ENE had a 74% higher mortality risk (HR = 1.74, 95% CI 1.09–2.78, *p* = 0.02) and those with unknown status of ENE also had an increased mortality risk (HR = 1.76, 95% CI 1.24—2.54, *p* = 0.002). Moreover, adjuvant chemotherapy was associated with lower mortality risk compared with no adjuvant chemotherapy (HR = 0.58, 95% CI 0.43–0.79, *p* = 0.001).

### 3.3. Cancer-Specific Survival

A total of 175 (57.6% of the cohort, 82.5% of mortality cases) patients in this cohort died of UC until December 2019. Patients with ENE had a non-significantly lower survival outcome than those without ENE (Figure 2, log-rank test *p* = 0.066). The 5-year CSS rates of the patients with ENE, unknown status of ENE, and without ENE were 22.9%, 32.8%, and 44.8%, respectively. In the univariable analysis, ENE status, age, surgical margin, tumor size, pathological T stage, multiple primaries, perioperative chemotherapy, and LND were significantly related to CSS (Table 2). The multivariable analysis revealed that patients with ENE had lower CSS than those without ENE (HR = 1.69, 95% CI 1.01–2.83, *p* = 0.045). In addition, greater LND, larger tumor size, older age, and no perioperative chemotherapy significantly reduced CSS. Adjuvant chemotherapy was associated with a reduced mortality risk by 43% compared with no adjuvant chemotherapy (HR = 0.57, 95% CI 0.40–0.80, *p* = 0.001). The survival curve of the bladder cancer patients stratified by peri-operative chemotherapy was shown in Figure 3 and Appendix A.

### 3.4. Association between Perioperative Chemotherapy and ENE Status

Among this cohort with pathological lymph node involvement, 54 (17.8%), 129 (42.4%), and 20 (6.6%) patients received neoadjuvant, adjuvant, and neoadjuvant plus adjuvant chemotherapy, respectively. However, 101 (33.2%) patients did not receive any perioperative chemotherapy. Within the group of patients with ENE (*n* = 48), 10 (20.8%), 22 (45.8%), and 7 (14.6%) patients received neoadjuvant, adjuvant, and neoadjuvant plus adjuvant chemotherapy, respectively. Only nine (18.8%) patients did not receive perioperative chemotherapy. In other words, adjuvant chemotherapy was administered after the identification of ENE in 29 (60.4%) patients. Patients with ENE who did not receive perioperative chemotherapy exhibited significantly shorter survival duration than those who received perioperative chemotherapy (median survival: 12.26 vs. 20.49 months). In the patients with ENE, adjuvant chemotherapy was related to better OS (log-rank test, *p* = 0.014) and CSS (log-rank test, *p* = 0.276) compared with the subjects without chemotherapy (Appendix A). In addition, adjuvant chemotherapy may provide benefits in OS (log-rank test, *p* < 0.001) and CSS (log-rank test, *p* < 0.001) in the patients with unknown status of ENE. Nevertheless, adjuvant chemotherapy was not significantly associated with OS and CSS in the patients without ENE. Neoadjuvant plus adjuvant chemotherapy provided similar impact on OS and CSS compared with adjuvant chemotherapy alone in bladder cancer patients with positive or unknown status of ENE (Appendix A).

## 4. Discussion

Positive surgical margins, higher pathological T stages, tumor sizes > 4 cm, and LND > 20% were associated with a higher rate of ENE of lymph nodes. In the multivariable analysis, ENE predicted poor OS and CSS. Notably, adjuvant chemotherapy seemed to correlate to the reduced risk of mortality in these patients.

ENE in metastatic lymph nodes is treated as a poor prognostic factor for many malignancies, including colon cancer, breast cancer, and prostate cancer [14,15]. ENE refers to the growth of metastatic cancer beyond the capsule of lymph nodes and into the adjacent tissues [5]. In the basic concepts of disease pathophysiology, the flow of the lymphatic fluid in collecting lymphatic vessels is affected by preload, afterload, and transmural pressure [16,17]. ENE of the metastatic nodes is considered a breach in the capsule of lymph nodes, resulting in the outspread of the metastatic clone to the nearby soft tissue [18,19]. Undoubtedly, this microscopic phenomenon would highly increase the risk of disease spread, ultimately affecting patients’ survival outcomes. However, there are no strict criteria for the microscopic diagnosis of ENE [20,21]. The detection of ENE may be influenced by observers, leading to a significant interobserver variability. A study by Fujii et al. considered ENE as the extracapsular growth of tumor cells, invasion of perinodal fat, or extranodal location of tumor cells [22]. Because of the emerging clinical significance of ENE, adopting a standard definition is important.

Using the database of the TCR, we demonstrated that ENE is an independent prognostic factor for patients with bladder cancer and lymph node metastasis who have undergone radical cystectomy and lymph node dissection. This result is similar to the results observed in previous studies. Both Fajkovic [8] and Fleischmann [5] stated that extracapsular extension of lymph nodes significantly worsened patient outcomes. By contrast, Kassouf enrolled 150 patients with bladder cancer with pN+M0 disease who had undergone radical cystectomy from 1993 to 2003 and observed that ENE was not an independent prognostic factor [23]. This conflicting result could be attributable to the fact that the majority (70%) of their patients received adjuvant chemotherapy. Therefore, adjuvant chemotherapy may interfere with results and mask the prognostic significance of ENE in bladder cancer patients who develop lymph node metastasis.

Neoadjuvant chemotherapy with platinum-based regimens has been proven to statistically and moderately improve survival outcomes among patients with MIBC [24,25]. However, the application of neoadjuvant chemotherapy is still limited by several factors, including patients’ performance status, renal function, age, and preference. For example, only 24.3% of the patients with nodal bladder cancer received neoadjuvant chemotherapy in Taiwan. Adjuvant chemotherapy might be a suitable option to compensate for the insufficiency of treatment before radical cystectomy and could be suggested for patients with a high risk of disease recurrence and progression. Lymph node involvement in cancer is a surrogate for distant metastasis and should be managed carefully.

Several series have reported that adjuvant chemotherapy after radical cystectomy can improve outcomes among patients with bladder cancer who have locally advanced disease [26,27]. The National Comprehensive Cancer Network guidelines (version 6.2020) of bladder cancer also state that adjuvant chemotherapy can be considered for patients in the pT3-4N0-3M0 stage [28,29]. In our cohort, we also identified a significant survival benefit of adjuvant chemotherapy in patients with nodal metastatic bladder cancer. Because ENE is a poor prognostic factor for patients with nodal metastatic bladder cancer, adjuvant chemotherapy can be suggested for such patients. As expected, the survival benefit of adjuvant chemotherapy was more significant among patients with ENE.

In our study, multiple primaries and BMI were also recorded in both univariable and multivariable analyses. Multiple primaries are defined as the presence of more than one UC lesion beyond the bladder, known as synchronous UC. Yousem conducted a retrospective analysis of 645 patients with a histologically confirmed diagnosis of UC of the bladder, ureter, or kidney. He concluded that synchronous UC was found in 2.3% of patients with bladder UC, 39% of those with ureteral UC, and 24% of those with renal UC [30]. In our study, we observed that multiple primaries of UC did not lead to a deterioration in OS and CSS. Low BMI was reported to be associated with a poor prognosis of bladder UC owing to the relatively poor nutrition status of the patients. However, Westhoff conducted a systematic review and found no association between BMI and the risk of progression of MIBC [31].

Our study has several limitations. First, this was a retrospective analysis; however, selection bias may have been partially mitigated by the nationwide registry system. Second, although ENE is the mandatory parameter to be collected in the registry system, not all pathologists reported ENE as a standard. The large number with unknown status of ENE limited the statistical power to identify the minor difference. Third, interobserver variability with regard to ENE could not be avoided because of the nature of the study, which involved collection of information from several institutions. Fourth, the sample size used in the statistical analysis for OS and CSS was not enough to explore the minor differences. Therefore, the conclusion based on this analysis should be validated in a large cohort. Finally, the lack of information on performance status and comorbidity index may bias the decision to use chemotherapy and influence the analysis between chemotherapy and outcomes.

## 5. Conclusions

In conclusion, ENE of lymph nodes significantly reduced both OS and CSS in patients with nodal bladder cancer who were treated with radical cystectomy and lymph node dissection. Adjuvant chemotherapy was associated with improved survival outcomes in bladder cancer patients with ENE.

## Figures and Tables

**Figure 1 cancers-13-04108-f001:**
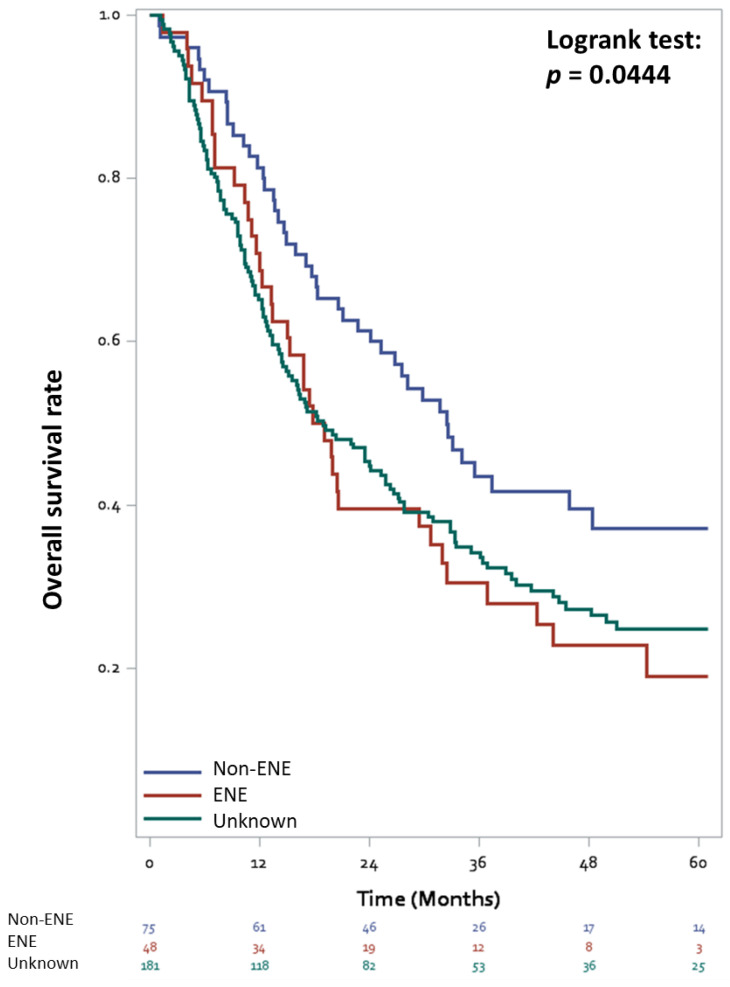
Overall survival curves stratified by the status of extranodal extension in bladder cancer patients with lymph node involvement treated with radical cystectomy.

**Figure 2 cancers-13-04108-f002:**
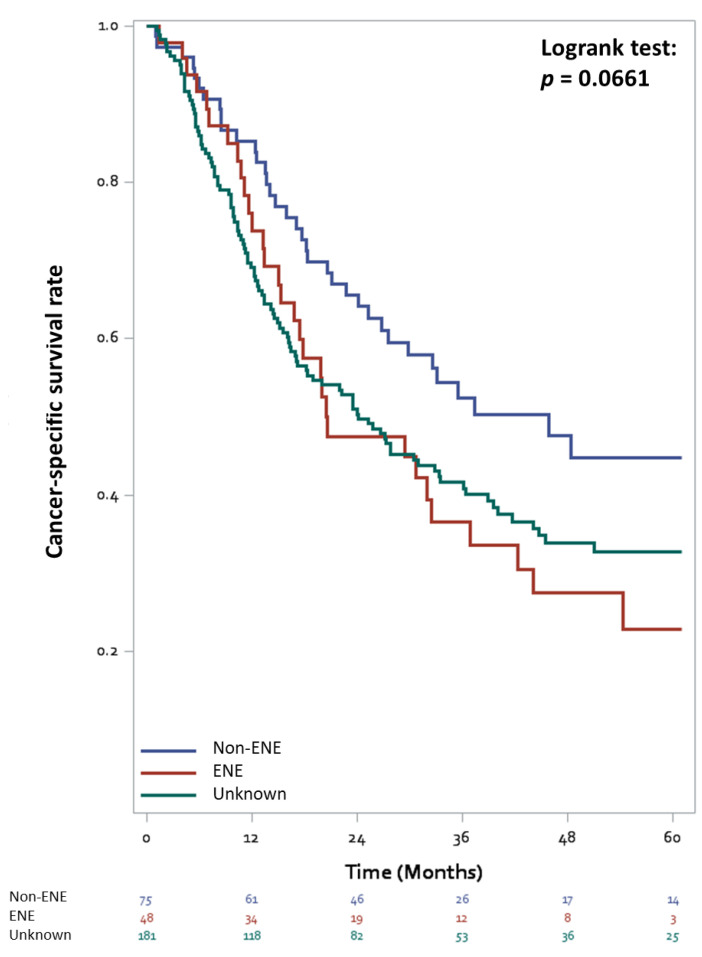
Cancer-specific survival curves stratified by the status of extranodal extension in bladder cancer patients with lymph node involvement treated with radical cystectomy.

**Figure 3 cancers-13-04108-f003:**
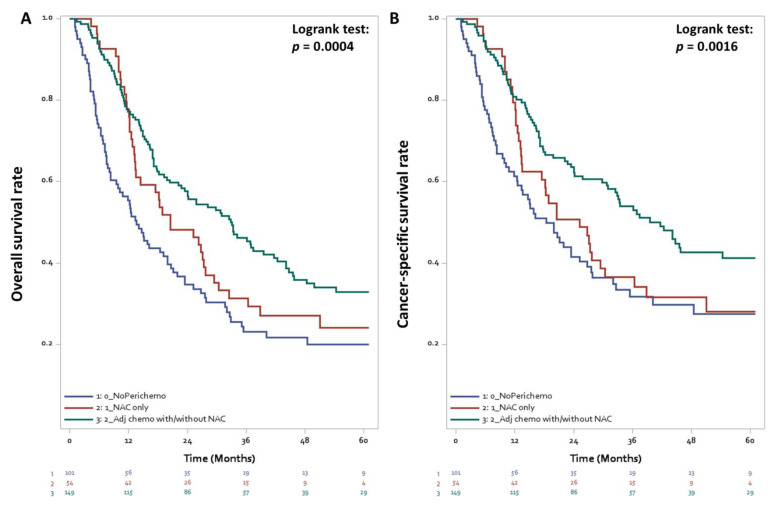
Overall survival (**A**) and cancer-specific survival (**B**) curves stratified by peri-operative chemotherapy in bladder cancer patients with lymph node metastases treated with radical cystectomy.

**Table 1 cancers-13-04108-t001:** Demographics of bladder cancer patients with lymph node involvement who received radical cystectomy.

Variables	Case Number	Extranodal Extension			*p* Value *
		Negative	Positive	Unknown	
All	304	75 (24.7%)	48 (15.8%)	181 (59.5%)	
Age (years)					0.98
25–64	136 (44.7%)	35 (46.7%)	22 (45.8%)	79 (43.6%)	
65–74	111 (36.5%)	24 (32.0%)	15 (31.3%)	72 (39.8%)	
≥75	57 (18.8%)	16 (21.3%)	11 (22.9%)	30 (16.6%)	
Gender					0.6
male	214 (70.4%)	53 (70.7%)	36 (75%)	125 (69.1%)	
female	90 (29.6%)	22 (29.3%)	12 (25%)	56 (30.9%)	
Tumor size (cm)					0.44
<4.0	88 (28.9%)	25 (33.3%)	11 (22.9%)	52 (28.7%)	
≥4.0	155 (51%)	39 (52%)	30 (62.5%)	86 (47.5%)	
Unknown	61 (20.1%)	11 (14.7%)	7 (14.6%)	43 (23.8%)	
Tumor grade					0.21
Low	3 (1%)	0	1 (2.1%)	2 (1.1%)	
High	301 (99%)	75 (100%)	47 (97.9%)	179 (98.9%)	
pT stage					0.73
T0	9 (3.0%)	1 (1.3%)	1 (2.1%)	7 (3.9%)	
Tis/a/1	6 (2.0%)	1 (1.3%)	0	5 (2.8%)	
T2	47 (15.5%)	12 (16.0%)	5 (10.4%)	30 (16.6%)	
T3	156 (51.3%)	40 (53.3%)	24 (50.0%)	92 (50.0%)	
T4	86 (28.3%)	21 (28.0%)	18 (37.5%)	47 (26.0%)	
pN stage					< 0.01
N1	112 (36.8%)	31 (41.3%)	8 (16.7%)	73 (40.3%)	
N2	156 (51.3%)	38 (50.7%)	29 (60.4%)	89 (49.2%)	
N3	36 (11.8%)	6 (8%)	11 (22.9%)	19 (10.5%)	
Specimen margin					0.18
Free	247 (81.3%)	63 (84%)	37 (77.1%)	147 (81.2%)	
Not free	50 (16.4%)	12 (16%)	9 (18.8%)	29 (16%)	
Unknown	7 (2.3%)	0	2 (4.2%)	5 (2.8%)	
Lymph node density					0.043
Unknown	14 (4.6%)	2 (2.7%)	1 (2.1%)	11 (6.1%)	
<20%	148 (48.7%)	45 (60.0%)	18 (37.5%)	85 (47%)	
≥20%	142 (46.7%)	28 (37.3%)	29 (60.4%)	85 (47%)	
Multiple primaries					< 0.01
Single	248 (81.6%)	54 (72%)	45 (93.8%)	149 (82.3%)	
Multiple	56 (18.4%)	21 (28%)	3 (6.2%)	32 (17.7%)	
Smoking history					0.67
Never	188 (61.8%)	44 (58.7%)	26 (54.2%)	118 (65.2%)	
Current	67 (22%)	16 (21.3%)	14 (29.1%)	37 (20.4%)	
Ever	46 (15.1%)	14 (18.7%)	8 (16.7%)	24 (13.3%)	
Unknown	3 (1%)	1 (1.3%)	0	2 (1.1%)	
Body mass index					0.17
<18	54 (17.8%)	20 (26.7%)	5 (10.4%)	29 (16%)	
18–23.9	121 (39.8%)	29 (38.7%)	22 (45.8%)	70 (38.7%)	
24–26.9	77 (25.3%)	18 (24%)	13 (27.1%)	46 (25.4%)	
≥27	52 (17.1%)	8 (10.7%)	8 (16.7%)	36 (19.9%)	
Peri-operativechemotherapy					0.02
No	101 (33.2%)	32 (42.7%)	9 (18.8%)	60 (33.1%)	
NAC alone	54 (17.8%)	8 (10.7%)	10 (20.8%)	36 (19.9%)	
AC alone	129 (42.4%)	31 (41.3%)	22 (45.8%)	76 (42%)	
NAC plus AC	20 (6.5%)	4 (5.3%)	7 (14.6%)	9 (5.0%)	

NAC = neoadjuvant chemotherapy; AC = adjuvant chemotherapy. * *p* value between the patients with and without extranodal extension.

**Table 2 cancers-13-04108-t002:** The Cox proportional hazard models of overall and cancer-specific survival in bladder cancer patients with lymph node involvement who received radical cystectomy.

Variables	Case Number	All Cause Death	Cancer Specific Death	Univariable Analysis						Multivariable Analysis					
				Overall Survival			Cancer Specific Survival			Overall Survival			Cancer Specific Survival		
				HR	95% CI	*p* Value	HR	95% CI	*p* Value	HR	95% CI	*p* Value	HR	95%CI	*p* Value
Age (years)															
25–64	136	79	69	1	(ref)		1	(ref)		1	(ref)		1	(ref)	
65–74	111	84	68	1.55	1.14–2.11	0.005	1.43	1.02–2.0	0.04	1.47	1.08–2.02	0.02	1.36	0.96–1.92	0.08
≥75	57	49	38	2.4	1.66–3.42	<0.001	2.11	1.4–3.13	<0.001	2.19	1.48–3.21	<0.001	1.87	1.21–2.85	0.004
Gender															
Male	214	150	120	1	(ref)		1	(ref)		-	-	-	-	-	-
Female	90	62	55	0.98	0.72–1.31	0.89	1.08	0.78–1.48	0.62	-	-	-	-	-	-
Tumor size (cm)															
<4	88	54	42	1	(ref)		1	(ref)		1	(ref)		1	(ref)	
≥4	155	115	100	1.55	1.13–2.16	0.008	1.73	1.22–2.51	0.003	1.57	1.13–2.20	0.01	1.73	1.20–2.53	0.004
Unknown	61	43	33	1.3	0.86–1.93	0.21	1.28	0.81–2.01	0.29	1.26	0.80–1.98	0.32	1.09	0.65–1.82	0.74
Tumor grade															
Low	3	2	2	1.13	0.19–3.53	0.87	1.34	0.22–4.21	0.68	-	-	-	-	-	-
High	301	210	173	1	(ref)		1	(ref)		-	-	-	-	-	-
Pathological T stage															
T0	9	4	4	1	(ref)		1	(ref)		1	(ref)		1	(ref)	
Tis/a/1	6	4	3	1.55	0.37–6.57	0.53	1.17	0.23–5.32	0.83	2.68	0.56–12.8	0.20	2.08	0.36–10.8	0.38
T2	47	24	19	1.13	0.44–3.84	0.82	0.90	0.34–3.09	0.84	1.58	0.53–5.90	0.45	1.44	0.46–5.54	0.56
T3	156	109	94	1.96	0.82–6.38	0.19	1.69	0.71–5.52	0.30	2.80	0.98–10.2	0.08	2.65	0.90–9.85	0.10
T4	86	71	55	2.71	1.12–8.89	0.05	2.10	0.86–6.93	0.15	3.69	1.28–13.6	0.03	3.12	1.04–11.7	0.05
Surgical margin															
Free	247	164	136	1	(ref)		1	(ref)					-	-	-
Not free	50	42	34	1.66	1.16–2.30	0.004	1.61	1.09–2.32	0.01				-	-	-
Unknown	7	6	5	2.03	0.8–4.19	0.09	2.05	0.72–4.5	0.12				-	-	-
Lymph node density															
<20%	148	88	67	1	(ref)		1	(ref)		1	(ref)		1	(ref)	
≥20%	142	114	98	1.83	1.38–2.43	< 0.001	2.06	1.51–2.82	< 0.001	1.47	1.10–1.97	0.01	1.69	1.22–2.35	0.002
Unknown	14	10	10	1.63	0.79–2.98	0.14	2.14	1.04–3.97	0.02	2.16	0.97–4.31	0.05	2.91	1.28–5.94	0.01
Extranodal extension															
No	75	44	36	1	(ref)		1	(ref)		1	(ref)		1	(ref)	
Yes	48	37	31	1.59	1.02–2.46	0.038	1.62	1.0–2.63	0.048	1.74	1.09–2.78	0.02	1.69	1.01–2.83	0.045
Unknown	181	131	108	1.5	1.07–2.13	0.02	1.51	1.05–2.23	0.03	1.76 **	1.24–2.54	0.001	1.76 **	1.20–2.64	0.01
Multiple primaries															
No	248	178	151	1	(ref)		1	(ref)		-	-	-	-	-	-
Yes	56	34	24	0.76	0.52–1.08	0.14	0.63	0.4–0.95	0.04	-	-	-	-	-	-
Smoking history															
Never	188	130	109	1	(ref)		1	(ref)		-	-	-	-	-	-
Current	67	45	39	0.97	0.68–1.34	0.84	1	0.68–1.43	1	-	-	-	-	-	-
Ever	46	36	26	1.47	1–2.10	0.04	1.26	0.81–1.91	0.28	-	-	-	-	-	-
Unknown	3	1	1	0.34	0.02–1.52	0.28	0.41	0.02–1.84	0.38	-	-	-	-	-	-
Body mass index															
<18	54	34	26	0.72	0.48–1.06	0.11	0.68	0.43–1.05	0.09	-	-	-	-	-	-
18–23.9	121	91	74	1	(ref)		1	(ref)		-	-	-	-	-	-
24–26.9	77	54	47	0.87	0.62–1.22	0.43	0.93	0.64–1.34	0.71	-	-	-	-	-	-
≥27	52	33	28	0.71	0.47–1.04	0.09	0.74	0.47–1.13	0.17	-	-	-	-	-	-
Neoadjuvant chemotherapy *															
No	230	160	128	1	(ref)		1	(ref)		1	(ref)		1	(ref)	
Yes	74	52	47	0.91	0.66–1.23	0.54	1.02	0.73–1.42	0.89	0.72	0.49–1.03	0.08	0.86	0.57–1.27	0.44
Adjuvant chemotherapy *															
No	155	118	99	1	(ref)		1	(ref)		1	(ref)		1	(ref)	
Yes	149	94	76	0.62	0.47–0.81	<0.001	0.60	0.44–0.80	<0.001	0.58	0.43–0.79	<0.001	0.57	0.40–0.80	0.001

CI = confidence interval; HR = hazard ratio; ref = reference; NAC = neoadjuvant chemotherapy; AC = adjuvant chemotherapy. * The models containing four perioperative chemotherapy categories, no chemotherapy, neoadjuvant chemotherapy alone, adjuvant chemotherapy alone, and neoadjuvant plus adjuvant chemotherapy, would be shown in the Appendix A. ** In the consideration of unknown status of extranodal extension (ENE) as reference, positive ENE was not associated with poorer overall (HR = 0.97, 95% CI 0.66–1.41) and cancer-specific survival (HR = 0.94, 95% CI 0.61–1.41).

## Data Availability

Data are available on request due to restrictions of privacy considerations of Taiwan Cancer Registry Database. The data presented in this study are available on request from the corresponding author. The data are not publicly available due to the consideration of patients’ privacy.

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
