# Peer review of "Extranodal Extension Predicts Poor Survival Outcomes among Patients with Bladder Cancer"

_cancers, 2021, doi:10.3390/cancers13164108_

Round 1

Reviewer 1 Report

The authors have well responded to my comments. 

Author Response

Thanks for your great review.

Reviewer 2 Report

The revised manuscript includes amendment of a number of minor errors, and the separation of cases into individual rather than grouped T stages will be useful for readers with specific expertise or interests in bladder cancer pathology. Although some English language editing will still be required to refine word usage, changes will be minor and would not affect the content or findings of the study. Some of the minor comments and corrections are listed below. Overall, I feel that the manuscript is suitable for publication.

  1. Improvement of English language in the Simple Summary highlights the issues with language in the Abstract that follows it, in particular the first 2 sentences. (Suggested amendments for these 2 sentences: Several lymph node-related prognosticators have been reported in bladder cancer patients with lymph node involvement who undergo radical cystectomy. However, the role of extranodal extension (ENE) remains debatable for outcome prediction.) Final English language polishing prior to publication would remove these grammatical or syntax errors, as there are too many to list.
  2. The word “deteriorated” is a passive, not an active verb. Where it is used in the manuscript, terms such as “reduced” (OSS / CSS) or “worsened” (patient outcomes) would be more suitable.
  3. The investigators have used both univariate/multivariate and univariable/multivariable (analysis) in their manuscript. There is debate about the usage of these terms, however this should just be made consistent. (https://bmjopen.bmj.com/content/bmjopen/8/5/e021129/DC2/embed/inline-supplementary-material-2.pdf?download=true)
  4. Line 280: Forth should be “fourth”.
  5. Line 313: “Neither he nor the other authors” should be corrected.

Author Response

Thank you for allowing us to modify and thereby improve the manuscript. We have made point-to-point revisions or statements per the reviewer’s comments. All revisions in the manuscript are marked up with the “track change” function for reference. We appreciate your and the referees’ efforts in helping us improve the quality of our paper, and hope this revision be published in Cancers finally.

Our responses to the critique are listed below:

Reviewer 1:

  1. Improvement of English language in the Simple Summary highlights the issues with language in the Abstract that follows it, in particular the first 2 sentences. (Suggested amendments for these 2 sentences: Several lymph node-related prognosticators have been reported in bladder cancer patients with lymph node involvement who undergo radical cystectomy. However, the role of extranodal extension (ENE) remains debatable for outcome prediction.) Final English language polishing prior to publication would remove these grammatical or syntax errors, as there are too many to list.

Response: Thanks for your careful review. We replaced the first two sentences in the Simple Summary with those you recommended.

  1. The word “deteriorated” is a passive, not an active verb. Where it is used in the manuscript, terms such as “reduced” (OSS / CSS) or “worsened” (patient outcomes) would be more suitable.

Response: Thanks for your wise comment. The word “deteriorated” was changed to either “reduced” (for OS/CSS) or “worsened” (for patient outcomes) in the abstract, on lines 61, 169, and 227.

  1. The investigators have used both univariate/multivariate and univariable/multivariable (analysis) in their manuscript. There is debate about the usage of these terms, however this should just be made consistent.

Response: We overlooked this point. We made this consistent by replacing multivariate with multivariable in the abstract and section materials and methods.

  1. Line 280: Forth should be “fourth”.

Response: The typo was corrected.

  1. Line 313: “Neither he nor the other authors” should be corrected.

Response: The statement was replaced with “The authors declared no conflict of interest.”

We trust that our manuscript can now be formally accepted for publication in the Journal with these changes. Please open the link for uploading the new version of the manuscript.

Respectfully Your,

This manuscript is a resubmission of an earlier submission. The following is a list of the peer review reports and author responses from that submission.

Round 1

Reviewer 1 Report

In this research, the authors use the National Taiwan Cancer Registry

to evaluate the prognostic impacts of extranodal extension (ENE) in patients with bladder cancer. This is an interesting issue. They concluded that bladder cancer patients with ENE have worse overall survival and cancer specific survival. I have some questions and comments.

  1. The major problem of this study is that ENE status is only available in 40% patients (133/304) and 60% data are unknown . The sample size (75 vs. 48) is too small to elucidate the impact of ENE in bladder cancer.
  2. In the Table 1.

  -Please add percentage of all items, for example 136 (44.7%) 111 (36.5%), and 57 (18.8%) for Age group.

  -Please add the pathologic nodal status (N1 or N2) and pathologic stage (III or IV)

  - P-value did not mean the difference between ENE+ and ENE- due to influence of unknown cases. To exclude the ENE unknown cases should be considered in this study.

  1. In the Table 2, patients with unknown ENE have the higher HR for OS and CSS than those with ENE+ in the multivariable analysis. Why ?
  2. Figure 1 is not cited in the manuscript. In the Results (3.1. Demographics, 3.3. Cancer-specific survival), 3.2... cannot be found.
  3. Your conclusion " Reduced OS and CSS outcomes were observed in the pathological nodal bladder cancer patients with ENE". But in the Figure 2, the p=0.661 is not significant. In the Table 2, patients with unknown ENE have the highest risk for overall and cancer death. 

6.Please analyze the impact of adjuvant chemotherapy on OS and CSS  according to ENE status using Kaplan–Meier analysis. 

Author Response

Thank you for giving us the opportunity to modify and thereby improve the manuscript. We have made point-to-point revisions or statements per the reviewer’s comments. All revisions in the manuscript are marked up with “track change” function for reference. We appreciate yours and the referees’ efforts in helping us improve the quality of our paper, and hope this revision be published in Cancers finally.

Our responses to the critique are listed below:

Reviewer 1:

  1. The major problem of this study is that ENE status is only available in 40% patients (133/304) and 60% data are unknown. The sample size (75 vs. 48) is too small to elucidate the impact of ENE in bladder cancer.

Response: Thanks for your wise comment. To investigate the role of ENE in the patients having radical cystectomy, we need to enroll the patients as more as possible. Taiwan Cancer Registry enrolled more than 97% of cancer patients in Taiwan and provided the standardized parameters for analysis. It is the best database in Taiwan but still has some limitations from the reporters. Although ENE is the mandatory parameter to be collected in the registry system, not all pathologists convinced and were willing to report ENE as a standard. Therefore, we can only analyze this important variable using the available data. Based on the statistical principle, the sample size decides the statistical power to identify the difference between groups. The small sample size would limit the identification of true difference. In contrast, if the difference of the specific variable can be found with satisfactory adjustment such as the multivariable analysis in this small sample cohort, its role will be confirmed. That is why we are comfortable to present the impact of ENE even the sample size seems not large. We have put this limitation in the section of Discussion.

  1. Table 1: (a) Please add percentage of all items, for example 136 (44.7%) 111 (36.5%), and 57 (18.8%) for Age group. (b) Please add the pathologic nodal status (N1 or N2) and pathologic stage (III or IV) (c) P-value did not mean the difference between ENE+ and ENE- due to influence of unknown cases. To exclude the ENE unknown cases should be considered in this study.

Response: Thanks for your comment to improve the comprehensiveness of our table. We added the percentage of all items, pathological nodal stage in Table 1. Because all patients (nodal bladder cancer) belonged to pathological stage IV, we did not add the pathological stage in Table 1.  The p value was recalculated by the exclusion of ENE unknown cases and the description was listed at the bottom of Table 1.

  1. In the Table 2, patients with unknown ENE have the higher HR for OS and CSS than those with ENE+ in the multivariable analysis. Why?

Response: In the Cox regression model, the reference of ENE variable was ENE negative group. Therefore, both unknown ENE and ENE positive had higher HR for OS and CSS than ENE negative group. We also compared unknown ENE and ENE positive directly and found no difference of HR for OS and CSS between them. That means patients with ENE negative have better survival outcomes compared with either ENE positive or unknown ENE. Besides, the mixture of ENE status in unknown ENE group may potentially tend to be ENE positive in this cohort.

  1. Figure 1 is not cited in the manuscript. In the Results (3.1. Demographics, 3.3. Cancer-specific survival), 3.2... cannot be found.

Response: Thanks for your reminder. It is our fault to lose this section at the editing phase. Please find the corresponding section (3.2. overall survival) with the citation of Figure 1 at the section 3.2 (Line 126).

  1. Your conclusion " Reduced OS and CSS outcomes were observed in the pathological nodal bladder cancer patients with ENE". But in the Figure 2, the p=0.661 is not significant. In the Table 2, patients with unknown ENE have the highest risk for overall and cancer death.

Response: Thanks for your view. We will precisely edit this sentence to “Reduced OS and CSS outcomes were observed in the pathological nodal bladder cancer patients with ENE compared with those without ENE”. Figure 2 showed the statistical analysis using log rank test in the univariable analysis, which was not well adjusted by other variables. Table 2 revealed the true impact of ENE status in the multivariable analysis. It was true unknown ENE showed higher risk of overall and cancer-specific death in the consideration of the reference as ENE negative. In fact, there was no difference between ENE positive and unknown ENE. In the consideration of unknown ENE as reference, positive ENE did not increase risk for overall (HR = 0.97, 95% CI 0.66-1.41) and cancer-specific survival (HR = 0.94, 95% CI. 0.61-1.41). (Added in Table 2)

  1. Please analyze the impact of adjuvant chemotherapy on OS and CSS according to ENE status using Kaplan–Meier analysis. 

Response: We added these two figures (Figure 3 a and 3b) to present the impact of adjuvant chemotherapy for improving outcomes. Hope these figures convince you.

Reviewer 2 Report

This straightforward manuscript has statistically shown in a well-annotated cohort of Taiwanese bladder cancer patients that extranodal extension was associated with poorer cancer-specific and overall survival. Although this result, and the additionally confirmed benefit of perioperative chemotherapy, might seem to be logical, studies such as this are critical for the development of standardised surgical techniques, pathology reporting and treatment regimens to maximise outcomes and improve analysis of novel treatments for advanced bladder cancer. The manuscript is well-structured and well-written, English language is excellent and referencing is very good. Tables and figures are clear and the length of the manuscript is appropriate considering its content. I feel that the manuscript is appropriate for publication in this journal pending minor amendments listed below.

  1. The Abstract is misleading in that the size of the original number of bladder cancer patients receiving radical cystectomy and bilateral pelvic lymph node dissection (1,303 patients) is stated, however it is not mentioned that the entire study is based on 304 of these patients where lymph node involvement was recorded (and only 123 where extranodal extension (positive or negative) was included in the pathology report). As such, results subsequently presented in the Abstract are misleading. The size of the cohort upon which the statistical analyses are based should be mentioned in the Abstract (it is not necessary to completely describe the division of the cohort).
  2. In the first paragraph of the Introduction section, the authors have used statistics from the early 2000s. More up-to-date statistics and references should be used to describe bladder cancer incidence and outcomes.
  3. I can’t seem to find a section 3.2.
  4. Although results are very clearly presented, it is noted that the cohort of patients included in many of the statistical analyses is quite small (75 negative / 48 positive for extranodal extension). This is because extranodal extension was unknown for the majority (181) of the 304 cases with lymph node involvement of their bladder cancer. I feel that this is not always obvious within the extensive statistical analyses, especially in the Results commentary, where percentages are used. I do not feel that major changes in the text are necessary as this would interrupt the flow of the manuscript. However, the low numbers of cases able to be included in statistical analyses and upon which the major conclusions of the authors are based is a limitation of the study and I feel that it should be added to the limitations listed in the final paragraph of the Discussion section.

Author Response

Thank you for giving us the opportunity to modify and thereby improve the manuscript. We have made point-to-point revisions or statements per the reviewer’s comments. All revisions in the manuscript are marked up with “track change” function for reference. We appreciate yours and the referees’ efforts in helping us improve the quality of our paper, and hope this revision be published in Cancers finally.

Our responses to the critique are listed below:

Reviewer 2:

Thanks for your nice review.

  1. The Abstract is misleading in that the size of the original number of bladder cancer patients receiving radical cystectomy and bilateral pelvic lymph node dissection (1,303 patients) is stated, however it is not mentioned that the entire study is based on 304 of these patients where lymph node involvement was recorded (and only 123 where extranodal extension (positive or negative) was included in the pathology report). As such, results subsequently presented in the Abstract are misleading. The size of the cohort upon which the statistical analyses are based should be mentioned in the Abstract (it is not necessary to completely describe the division of the cohort).

Response: Thanks for your reminder. We add “Based on the 304 patients with lymph node involvement, the presence of ENE…….. “in the abstract to reduce the misleading.

  1. In the first paragraph of the Introduction section, the authors have used statistics from the early 2000s. More up-to-date statistics and references should be used to describe bladder cancer incidence and outcomes.

Response: Thanks. We replaced the up-to-date statistics in the first paragraph.

  1. I can’t seem to find a section 3.2.

Response: Thanks for your reminder. It is our fault to lose this section at the editing phase. Please find the corresponding section (3.2. overall survival) with the citation of Figure 1 at the section 3.2 (Line 126).

  1. Although results are very clearly presented, it is noted that the cohort of patients included in many of the statistical analyses is quite small (75 negative / 48 positive for extranodal extension). This is because extranodal extension was unknown for the majority (181) of the 304 cases with lymph node involvement of their bladder cancer. I feel that this is not always obvious within the extensive statistical analyses, especially in the Results commentary, where percentages are used. I do not feel that major changes in the text are necessary as this would interrupt the flow of the manuscript. However, the low numbers of cases able to be included in statistical analyses and upon which the major conclusions of the authors are based is a limitation of the study and I feel that it should be added to the limitations listed in the final paragraph of the Discussion section.

Response: Thanks for your wise comment. As your suggestion, we added one more limitation in the section of Discussion to mention the low number of cases in our study and its impact on the statistical analysis. 

Round 2

Reviewer 1 Report

The authors have made significant revisions of their manuscript. There are a few more things:

1.Please respond to my comment 6, "Please analyze the impact of adjuvant chemotherapy on OS and CSS according to ENE status using Kaplan–Meier analysis".  The Figure 3 is analyzed  stratified by the use of peri-operative chemotherapy. 

2. The case number of neoadjuvant chemotherapy is 54 in the Table 1 but 74 int the Table 2.  Why?

Author Response

Thank you for giving us the opportunity to modify and thereby improve the manuscript. We have made point-to-point revisions or statements per the reviewer’s comments. All revisions in the manuscript are marked up with “track change” function for reference. We appreciate yours and the referees’ efforts in helping us improve the quality of our paper, and hope this revision be published in Cancers finally.

Our responses to the critique are listed below:

Reviewer 1:

  1. Please respond to my comment 6, "Please analyze the impact of adjuvant chemotherapy on OS and CSS according to ENE statususing Kaplan–Meier analysis".  The Figure 3 is analyzed  stratified by the use of peri-operative chemotherapy. .

Response: Thanks! Based on your comment, we added the description about the impact of adjuvant chemotherapy on OS and CSS according to ENE status using K-M method in section 3.4 and supplementary Figure 1. In the patients with positive ENE, adjuvant chemotherapy improved OS (log-rank test, p = 0.014) and CSS (log-rank test, p = 0.257) compared with the subjects without chemotherapy. Besides, adjuvant chemotherapy also provided benefit on OS (log-rank test, p < 0.001) and CSS (log-rank test, p < 0.001) in the patients with unknown status of ENE. Nevertheless, adjuvant chemotherapy did not significantly improve OS and CSS for the patients with negative ENE.

  1. The case number of neoadjuvant chemotherapy is 54 in the Table 1 but 74 int the Table 2.  Why?

Response: Thanks for your detailed check. The case number of patients having neoadjuvant chemotherapy is 54. That is our error to mistype 74 in Table 2. We have corrected it in the revised version.

We trust that with these changes, our manuscript can now be formally accepted for publication in the Journal

Respectfully Your,
